# Zero-shot image classification based on class representation learning and attribute embedding learning

Huabo Shen[1,2], Xiaodong Sun[2,3], Youmin Hu[4]*, Changgeng Li[2,5]*, Qinmiao Zhu[1,6], Qin Li[7]*

1 School of Mechanical Science and Engineering, Huazhong University of Science and Technology, Wuhan, China, 2 CRRC Qingdao Sifang Co., Ltd., Qingdao, China, 3 School of Mechanical, Electronic and Control Engineering, Beijing Jiaotong University, Beijing, China, 4 Wuxi Research Institute, Huazhong University of Science and Technology, Wuxi, China, 5 College of Information Science and Technology, Qingdao University of Science and Technology, Qingdao, China, 6 Wuhan Digital Design and Manufacturing Innovation Center Co., Ltd., Wuhan, China, 7 School of Computer and Software Engineering, Shenzhen Institute of Information Technology, Shenzhen, China

* 64881623@qq.com (YH); ChanggengLi@qust.edu.cn (CL); liqin@sziit.edu.cn (QL)

## Abstract

Zero-shot learning (ZSL) aims to classify unseen classes by leveraging semantic information from seen classes, addressing the challenge of limited labeled data. In recent years, ZSL methods have focused on extracting attribute-level features from images and aligning them with semantic features within an embedding space. However, existing approaches often fail to account for significant visual variations within the same attribute, leading to noisy attribute-level features that degrade classification performance.To tackle these challenges, we propose a novel zero-shot image classification method named CRAE (Class Representation and Attribute Embedding), which combines class representation learning and attribute embedding learning to enhance classification robustness and accuracy. Specifically, we design an adaptive softmax activation function to normalize attribute feature maps, effectively reducing noise and improving the discriminability of attribute-level features. Additionally, we introduce attribute-level contrastive learning with hard sample selection to optimize the attribute embedding space, reinforcing the distinctiveness of attribute representations. To further increase classification accuracy, we incorporate class-level contrastive learning to enhance the separation between features of different classes. We evaluate the effectiveness of our approach on three widely used benchmark datasets (CUB, SUN, and AWA2), and the experimental results demonstrate that CRAE significantly outperforms existing state-of-the-art methods, proving its superior capability in zero-shot image classification.

**Data availability statement:** The datasets referenced in the text—CUB-200-2011 (CUB), SUN, and Animals with Attributes 2 (AWA2)—can be accessed through the following URLs: The CUB-200-2011 dataset is available at http://www.vision.caltech.edu/datasets/cub_200_2011/, the SUN dataset can be found at http://groups.csail.mit.edu/vision/SUN/, and the Animals with Attributes 2 (AWA2) dataset is hosted at http://attributes.kyb.tuebingen.mpg.de/.

**Funding:** This work is supported by Natural Science Foundation of Guangdong Province under Grant 2023A1515011845 and Shenzhen Science and Technology Major Special Project under Grant KJZD20231023092602004). The funders had no role in study design, data collection and analysis, decision to publish, or preparation of the manuscript.

**Competing interests:** The authors have declared that no competing interests exist.

## Introduction

With the rapid advancement of deep learning technologies, supervised image classification models have achieved increasingly higher classification accuracies, even surpassing human-level performance. However, these models heavily rely on large-scale labeled datasets and are inherently limited to classifying only seen classes. The high cost of manually labeled data and the inevitable occurrence of unseen classes in real-world applications significantly restrict the practical deployment of supervised image classification models. To address this limitation, **Zero-Shot Learning (ZSL)** has emerged as a promising paradigm that leverages semantic auxiliary information shared between seen and unseen classes to enable knowledge transfer from seen classes to unseen classes.

Zero-shot image classification aims to identify unseen classes without any direct labeled training data, relying solely on semantic relationships learned from seen classes. Depending on the testing scenario, zero-shot learning can be categorized into **Conventional Zero-Shot Learning (CZSL)**, where only unseen classes are tested, and **Generalized Zero-Shot Learning (GZSL)**, where both seen and unseen classes are tested simultaneously.

To tackle the challenges inherent in classifying unseen classes, zero-shot learning methods can be broadly categorized into two main approaches: **embedding-based** and **generative-based**. Embedding-based zero-shot learning methods aim to map visual features and semantic information into a shared embedding space, enabling the alignment of image features with class semantic vectors. However, these methods often overlook the diversity of local visual features corresponding to the same attribute, leading to noise contamination and reduced classification accuracy. On the other hand, generative-based methods synthesize features for unseen classes using generative models, effectively transforming the zero-shot classification problem into a supervised learning task. Although generative methods alleviate class bias, they often face challenges in generating high-quality and discriminative features, which can limit classification performance.

Recently, attention-based zero-shot learning methods have shown promising results by leveraging attribute descriptions to learn localized features that are more semantically relevant. Despite these advancements, existing methods still struggle with visual representation variability and noise interference in attribute features, which hinder accurate image classification. Therefore, it is essential to develop a method that can effectively capture both **class-level and attribute-level features** while minimizing noise and preserving discriminability.

In light of the above challenges, we propose a novel method for zero-shot image classification named **CRAE (Class Representation and Attribute Embedding)**, which effectively combines class representation learning and attribute embedding learning. The overall architecture of the proposed framework is illustrated in Fig 1.

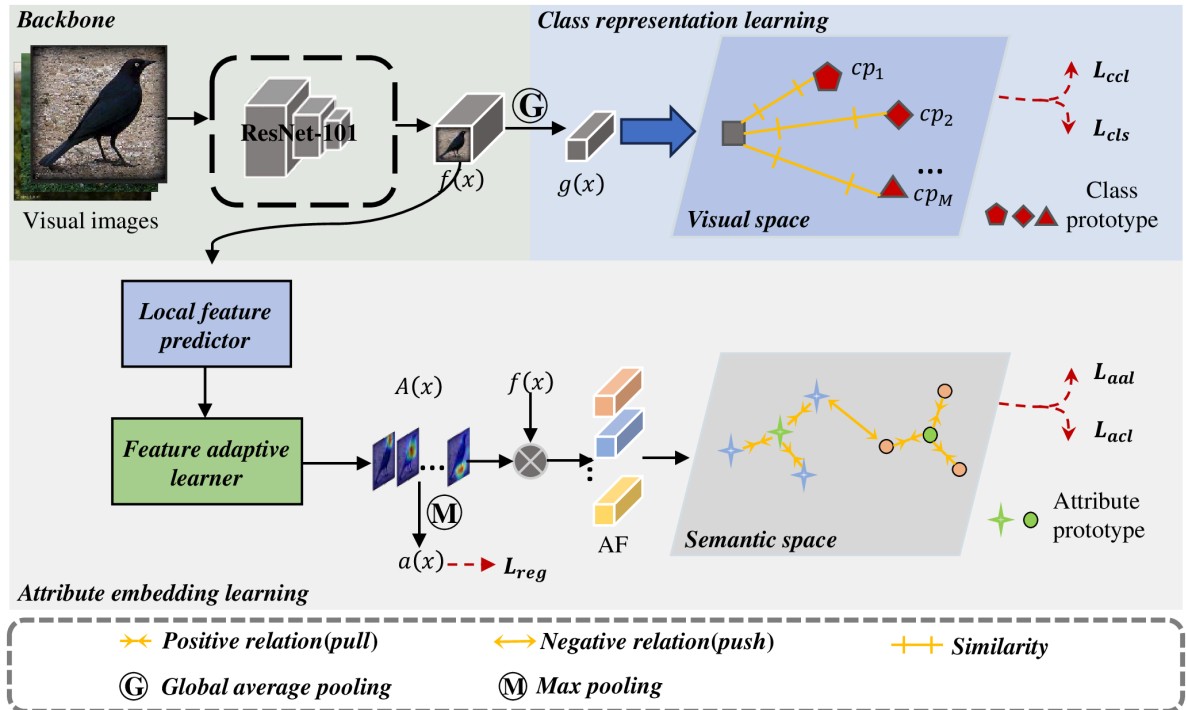

**Fig 1.** **Illustration of our proposed framework, which contains two parts: class representation learning and attribute embedding learning.** Class representation learning includes global feature alignment and class-level contrastive learning. Attribute embedding learning includes attribute localization, attribute feature extraction, attribute prototype alignment, and attribute-level contrastive learning.

Our contributions are summarized as follows:

- We introduce an innovative **attribute-based adaptive softmax activation function**, which dynamically normalizes attribute feature maps, effectively reducing noise and enhancing the discriminability of attribute-level features through a weighted sum of local features.
- We develop a novel **attribute-level contrastive learning framework** that leverages hard sample selection to constrain and refine attribute-level features. This significantly enhances the discriminability of the attribute embedding space. Furthermore, by integrating class-level contrastive loss, we optimize global feature representations, leading to a substantial improvement in model performance.
- Our method seamlessly integrates **class representation learning with attribute embedding learning**, facilitating a more robust semantic transfer between seen and unseen classes. Experiments conducted on three highly challenging benchmark datasets, i.e., CUB [1], AWA2 [2] and SUN [3], demonstrate that our approach consistently outperforms the latest state-of-the-art ZSL methods, highlighting its superior capability in addressing zero-shot image classification challenges.

The structure of the remaining part of this paper is as follows. The Related work section mainly introduces the related works of zero-shot learning and contrastive learning, followed by the details of our proposed method in the method section, where we present the basic notations, network architecture, loss functions, as well as the process of training and testing. We conduct extensive experiments on the proposed method in the Experiments section, and summarize the whole paper in the Conclusion section.

## Related work

In this section, we discuss the foundational work on zero-shot learning (ZSL) and contrastive learning, which are the primary techniques leveraged in our proposed approach.

### Zero-shot learning

Zero-shot learning has seen significant advancements, particularly in two main approaches: embedding-based methods and generative-based methods.

**Embedding-Based Methods**: Embedding-based ZSL methods typically learn a function that maps visual features to a semantic space, where these features are aligned with class semantic vectors [4,5]. These methods are generally end-to-end, allowing for a direct association between visual and semantic information. A common approach involves the use of convolutional neural networks (CNNs) to extract global visual features from images, which are then projected into a semantic space for alignment [6]. However, a key limitation of these methods is that global visual features often contain background information that is irrelevant to the semantics, which can widen the gap between visual and semantic features and ultimately degrade the performance of ZSL. To address this, recent methods have incorporated attention mechanisms to focus on more discriminative local visual features. For example, DAZLE [7] integrates attention mechanisms to acquire attribute-level visual features by emphasizing local visual features that are most relevant to each attribute. This method employs an attribute attention mechanism to prioritize attributes that are critical for classification. Similarly, APN [8] uses convolutional kernels specific to each attribute to extract feature maps, followed by max pooling to obtain attribute intensity, which is then aligned with class semantic vectors through a regression loss. Despite these advancements, some methods like APN overlook the variability in visual representation of the same attribute, which could lead to less robust performance.

**Generative-Based Methods**: On the other hand, generative-based ZSL methods generate visual features for unseen classes using generative models like GANs or VAEs [9–15]. These synthetic features are subsequently used to train a supervised classifier. While these methods can reduce bias towards seen classes, they are often not end-to-end, and the quality of generated features during the generation phase significantly impacts the final classification performance. Moreover, the training of generative models can be relatively unstable compared to embedding-based methods.

**Comparative Insight**: Compared to generative-based methods, embedding-based methods are generally more stable and intuitive in reflecting the relationship between visual and semantic modalities. The method proposed in this paper falls under the embedding-based category, where we enhance the approach by focusing on the extraction of discriminative attribute-level features with reduced noise.

### Contrastive learning

Contrastive learning is a technique that aims to learn discriminative representations by maximizing the similarity between positive sample pairs and minimizing the similarity between negative sample pairs. The effectiveness of contrastive learning heavily relies on the proper selection of positive and negative samples.

**Unsupervised Contrastive Learning**: In unsupervised tasks, methods like SimCLR [16] generate two augmented views of the same image and treat them as positive pairs, while other images serve as negative pairs. This approach has been effective in extracting discriminative features from images. MoCo [17] introduces momentum optimization and dictionary learning into contrastive learning, establishing an efficient self-supervised learning framework.

**Supervised Contrastive Learning**: In supervised tasks, supervised contrastive learning [18] extends this idea by treating images of the same class as positive samples and images of different classes as negative samples. This approach has been effectively integrated into the zero-shot learning domain. For instance, EMP applies supervised contrastive learning to address the domain shift problem by constraining attribute features across different domains.

**Recent Advances**: Recent methods have continued to refine contrastive learning techniques. CE-GZSL [19] introduces contrastive embedding, combining class-level and instance-level contrastive learning to leverage both forms of supervision for highly discriminative features. SCL [20] introduces a self-supervised contrastive learning mechanism that focuses on minimizing the distance between training samples and their augmented variants, thereby enhancing sample discrimination. EMP [21] treats different seen class samples of the same attribute as multiple source domains and different unseen class samples as multiple target domains. By introducing attribute-level contrastive learning, EMP constrains attribute-level features to improve cross-domain transferability.

**Application in Our Work**: Building on these insights, our approach extracts attribute-level features with reduced noise compared to previous methods. We further enhance the effectiveness of contrastive learning by designing attribute-level contrastive learning based on hard sample selection, which effectively reduces feature disparities in the embedding space. This refinement allows for better alignment of attribute features and improves overall zero-shot learning performance.

## Method

In this section, we first present the notation and problem settings for zero-shot learning. Subsequently, we delve into a detailed presentation of our proposed framework and formulation. To elaborate, our model aims to acquire highly discriminative visual features by jointly optimizing class representation learning and attribute embedding learning.

### Notation and problem settings

The goal of zero-shot learning is to transfer knowledge using class semantics to achieve classification of target classes. Seen classes ($\mathcal{Y}^s$) and unseen classes ($\mathcal{Y}^U$) are disjoint, i.e., $\mathcal{Y}^s \cap \mathcal{Y}^U = \varnothing$. The training set for zero-shot learning is represented as $T^s = \{x_i^s, y_i^s, \varphi(y_i^s)\}_{i=1}^{N^s}$. Where $x_i^s$ represents the image sample, $y_i^s$ is the corresponding class label, and $\varphi(y_i^s) \in R^K$ is the class semantic vector corresponding to that label. The test set of zero-shot learning can be represented as $T^u = \{x_i^u, y_i^u, \varphi(y_i^u)\}_{i=1}^{N^u}$. Class semantic vector is defined as $CS = \{\varphi(i)\}_{i=1}^M$. Here, $M$ is the number of categories. The objective of conventional zero-shot learning is to learn a mapping from the space of seen class images to the set of labels of unseen classes, i.e., $\mathcal{X}^S \rightarrow \mathcal{Y}^U$, thereby enabling classification of samples from unseen classes. The objective of generalized zero-shot learning is to learn a mapping from the space of seen class images to the set of all class labels, i.e., $\mathcal{X}^S \rightarrow \mathcal{Y}^U \cup \mathcal{Y}^s$, thus enabling classification of samples from both seen and unseen classes.

### Overview

Fig 1 shows our proposed framework, which contains two parts: class representation learning and attribute embedding learning, Class representation learning includes class prototype alignment and class-level contrastive learning. Attribute embedding learning includes attribute localization, attribute feature extraction, attribute prototype alignment, and attribute-level contrastive learning.

**Class prototype generator and attribute prototype generator.** The class prototype generator maps class semantics $CS = \{\varphi(i)\}_{i=1}^M$ to class prototypes $CP = \{cp_i\}_{i=1}^M$, and the attribute prototype generator maps attribute one-hot vectors $AS = \{as_j\}_{j=1}^K$ to attribute prototypes $AP = \{ap_j\}_{j=1}^K$. The generated high-dimensional class prototypes and attribute prototypes exhibit high sparsity, which is beneficial for improving performance.

**Class representation learning.** Given input image $x$, we utilize the ResNet-101 network to extract the local features $f(x)$ of the image. Then, after applying global average pooling, we obtain the global visual feature $g(x)$ of the image. Next, we introduce class-level contrastive learning to enhance the discriminative nature of the global visual feature $g(x)$. Finally, we align the global visual feature with the corresponding class prototype for image classification.

**Attribute embedding learning.** The local visual features $f(x)$ are processed through local feature predictor to obtain the original attribute feature maps $A(x)$, which are then normalized after passing through feature adaptive learner. For

each attribute feature map, the sum of values at each position is equal to 1. Weighting the local visual features by the relationship scores between attributes and local visual features in the attribute feature maps yields the attribute-level features $AF = \{af_i\}_{i=1}^{K}$. By constraining the attribute-level features through attribute-level contrastive learning, the discriminative ability of the attribute embedding space can be enhanced. Finally, aligning the attribute-level features with the corresponding attribute prototypes allows for distinguishing different attributes in the images, facilitating the transfer of knowledge to unseen classes.

## Class representation learning

**Class-level contrastive learning.** To enhance the ability of the visual embedding space to distinguish between different categories, we introduce class-level contrastive learning. Specifically, we treat features from the same class as positive samples and features from different classes as negative samples, aiming to maximize the similarity between positive samples and minimize the similarity between negative samples. For a batch of images, the class-level contrastive loss is formulated as:

$$\mathcal{L}_{ccl} = \frac{1}{B} \sum_{i=1}^{B} \frac{-1}{|P(i)|} \sum_{p \in P(i)} \log \frac{\exp(\cos(g(i), g(p))/\tau)}{\sum_{a=1, a \neq i}^{B} \exp(\cos(g(i), g(a))/\tau)} \tag{1}$$

where $P(i) \equiv \{p \in \{1, ..., B\} : y_p = y_i, p \neq i\}$ is the set of indices of all positives and $|P(i)|$ is its cardinality. Here, $\tau$ represents the temperature parameter.

By maximizing similarity among global features of samples within the same class and minimizing similarity between different classes, it strengthens coarse-grained category discrimination boundaries, improving the model's ability to differentiate visually similar categories.

**Global features alignment.** To accomplish the classification task, it is necessary to match the global visual features $g(x)$ of image samples to their corresponding class prototypes. We utilize a cross-entropy loss based on cosine similarity to constrain the global visual features and class prototypes. For a batch of images, the classification loss function is expressed as:

$$\mathcal{L}_{cls} = -\frac{1}{B} \sum_{i=1}^{B} \log \frac{\exp(\alpha \cdot \cos(g(x), cp_y))}{\sum_{i=1}^{M} \exp(\alpha \cdot \cos(g(x), cp_i))}, \tag{2}$$

where $\alpha$ is the scale factor, $B$ is the number of images in a batch, and $M$ is the number of classes. $\cos(\cdot, \cdot)$ denotes the cosine similarity between two feature vectors.

## Attribute embedding learning

**Attribute features extraction.** We use $K$ $1 \times 1 \times C$ convolutional kernels to transform the channel dimension of local visual features to $K$ dimensions, obtaining $K$ attribute feature maps, representing the relationship scores between each attribute and the local features. These attribute feature maps are normalized to [0,1] using $K$ attribute-based adaptive softmax activation functions, expressed as $A(x)_i = \text{Softmax}(A(x)_i * \rho_i)$. Here, $A(x)_i$ represents the feature map of the $i$-th attribute, $\rho_i$ represents the parameter to be learned, ranging from 0 to 1. This can be interpreted as applying a dynamic "temperature" scaling to each attribute's response distribution. Smaller $\rho_i$ values smooth the softmax outputs, suppressing noisy responses, whereas larger $\rho_i$ values amplify discriminative attributes. This mechanism enables the model to adaptively optimize attribute feature quality, reducing irrelevant interference and enhancing the representational robustness of

the attribute feature space. Attribute features are weighted sums of multiple local features, where local features that are closely related have larger weights. This process can be implemented using matrix multiplication. Specifically, the attribute feature maps are reshaped and transposed sequentially to make their dimensions $K \times H \times W$, while the local visual features are reshaped to have dimensions $H \times W \times C$. Finally, the two are multiplied using matrix multiplication to obtain the preliminary attribute features $AF = \{af_i\}_{i=1}^{K} \in R^{K \times C}$.

**Attribute localization.** According to [8], we take the maximum value of each attribute feature map as the strength value of the attribute. The predicted representation of all attributes of the image is denoted as $a(x)$. Then, we align $a(x)$ with the class semantics vector $CS$ using regression loss:

$$\mathcal{L}_{reg} = \|a(x) - \varphi(y)\|_2^2, \tag{3}$$

where $y$ is the label of image $x$. This loss is intended to improve the localization ability of attributes.

**Attribute prototype alignment.** To endow the model with the capability to differentiate between different attributes, we match the obtained attribute-level features with their corresponding attribute prototypes. Specifically, in the attribute embedding space, we bring the features of the same attribute closer to the corresponding attribute prototypes. Including attributes not present in the image in the alignment process would lead to negative optimization issues. Therefore, for a batch of images, we exclude attributes that are not present in the images to obtain the set of attribute-level features $\widehat{AF} = \{\widehat{af}_j\}_{j=1}^{\hat{K}}$. The attribute alignment loss is defined as follows:

$$\mathcal{L}_{aal} = \sum_{j=1}^{\hat{K}} \text{Relu}(\cos(\widehat{af}_j, ap_j) - 0.5 \min_{j' \neq j} \cos(\widehat{af}_j, ap_{j'})) \tag{4}$$

where Relu($\cdot$) denotes the recpdfied linear unit, preserving positive values, and $\hat{K}$ represents the number of feature points in a batch.

**Attribute-level contrastive learning.** In different samples, the same attribute often exhibits significant visual differences. This results in low similarity of the same attribute in the embedding space and even higher similarity between different attributes, severely affecting the ability of knowledge transfer. To address this issue, one approach is to bring visual features with the same attributes closer to each other, while keeping features with different attributes farther apart, thus enhancing the discriminability of attribute embeddings. Specifically, a straightforward method is to use supervised contrastive learning to constrain attribute-level features. However, supervised contrastive learning without removing simple samples can lead to increased computational costs and may result in blurry boundaries between different attributes. Therefore, we propose a supervised contrastive loss based on hard sample selection to constrain attribute-level features, aiming to learn discriminative attribute embedding spaces and improve the model's ability to extract attributes. Similar to the attribute prototype alignment discussed above, it is necessary to filter out attributes not present in the images to obtain a collection of attribute-level features for a batch of images, i.e., $\widehat{AF} = \{\widehat{af}_j\}_{j=1}^{\hat{K}}$, which will participate in attribute-level contrastive learning. The selection of positive and negative samples in the attribute-level contrastive loss based on hard sample selection is based on the sorting of class labels and cosine similarities. Specifically, for a certain attribute-level feature $\widehat{af}_j$, other attribute-level features with the same attribute label are all positive samples. The cosine similarity between these positive samples and $\widehat{af}_j$ is calculated and sorted in descending order. The positive samples at the front of the sorting are simple positive samples, while those at the back are hard positive samples. By excluding the top $\mu$ ($0 \leq \mu < 1$) simple positives from all positive samples, we obtain the set of hard positive samples. Similarly, attribute-level features with different attribute labels are all negative samples. The cosine similarity between these negative samples and $\widehat{af}_j$ is calculated and sorted in ascending order. The negative samples at the front of the sorting are simple negative samples, while those

at the back are hard negative samples. By excluding the top $\varepsilon$ ($0 \leq \varepsilon < 1$) simple negatives from all negative samples, we obtain the set of hard negative samples. The similarity between $\widehat{af_j}$ and $af_{ju}^+$ is defined as $S(\widehat{af_j}, af_{ju}^+) = \exp(\cos(\widehat{af_j}, af_{ju}^+)/\tau)$. The similarity between $\widehat{af_j}$ and $af_{jv}^-$ is defined as $S(\widehat{af_j}, af_{jv}^-) = \exp(\cos(\widehat{af_j}, af_{jv}^-)/\tau)$. Here, $\tau$ denotes a temperature parameter to control the degree of attention to hard negatives. Then, the attribute-level contrastive loss is represented as:

$$\mathcal{L}_{acl} = \frac{1}{\hat{K}} \sum_{j=1}^{\hat{K}} \left[ -\frac{1}{U} \sum_{u=1}^{U} \log \left( \frac{S(\widehat{af_j}, af_{ju}^+)}{\sum_{u=1}^{U} S(\widehat{af_j}, af_{ju}^+) + \sum_{v=1}^{V} S(\widehat{af_j}, af_{jv}^-)} \right) \right] \tag{5}$$

where $U$ and $V$ are the number of positives and negatives of $\widehat{af_j}$, respectively.

At a finer granularity, it utilizes hard positive and negative mining strategies to focus on challenging attribute feature pairs, reducing intra-attribute variation across samples and minimizing inter-attribute confusion. This effectively refines decision boundaries within the attribute embedding space.

## Optimization

The training of the entire framework is end-to-end. Combining the loss functions proposed in class representation learning and attribute embedding learning, the overall loss function of the model is represented as:

$$\mathcal{L} = \mathcal{L}_{cls} + \lambda_1 \mathcal{L}_{reg} + \lambda_2 \mathcal{L}_{aal} + \lambda_3 \mathcal{L}_{acl} + \lambda_4 \mathcal{L}_{ccl} \tag{6}$$

Where $\lambda_1$, $\lambda_2$, $\lambda_3$ and $\lambda_4$ are the loss coefficients for regression loss, attribute alignment loss, attribute-level contrastive loss, and class-level contrastive loss, respectively.

## Zero-shot classification

For conventional zero-shot learning tasks, given an image $x$, the expected prediction of the class prototype distance from the class features $g(x)$ is closest, the prediction formula is:

$$\hat{y} = \arg\max_{\bar{y} \in Y^u} \alpha \cdot \cos(g(x), cp_{\bar{y}}), \tag{7}$$

For generalized zero-shot learning tasks, the prediction formula involves reducing the cosine similarity between seen class prototypes and class features using a calibration factor $\gamma$, the prediction formula is:

$$\hat{y} = \arg\max_{\bar{y} \in Y^u \cup Y^s} \alpha \cdot \cos(g(x), cp_{\bar{y}}) - \gamma \mathbb{I}[\bar{y} \in Y^s], \tag{8}$$

Where $\alpha$ is the scaling factor and $\gamma$ is the calibration factor. When $\bar{y}$ represents an unseen class, it satisfies $\mathbb{I} = 1$; otherwise, $\mathbb{I} = 0$. During the training phase, the model updates its network parameters by minimizing the overall loss function in Equation 6. During the testing phase, it obtains predicted labels using Equation 7 and Equation 8 respectively. The complete process of the proposed algorithm is illustrated in Algorithm 1.

**Algorithm 1 CRAE.**

**Input:** The training set: $T^s = \{x_i^s, y_i^s, \varphi(y_i^s)\}_{i=1}^{N^s}$; Learning rate: $10^{-3}$; Attribute semantics: $K$ one-hot vectors; The hyper-parameters: $\lambda_1$, $\lambda_2$, $\lambda_3$, $\lambda_4$, $\mu$ and $\varepsilon$; The temperature parameter: $\tau$; The maximal number of training epochs: $T$.

**Output:** Labels for the testing set.

 // **Training stage:**

 1: **for** $epoch = 1$ to $T$ **do**

 2: Use ResNet101 to obtain the local visual feature $f(x)$ and the global visual feature $g(x)$;

 3: Use attribute prototype generator and class prototype generator to obtain attribute prototypes $AP$ and class prototypes $CP$;

 4: Use convolutional neural networks to obtain attribute prediction values $a(x)$ and attribute-level features $AF$;

 5: Compute the loss based on Equation 6 and update the parameters of CRAE through backpropagation;

 6: **end for**

 // **Testing stage:**

 1: For the CZSL setting, compute the labels for the testing set via Equation 7;

 2: For the GZSL setting, compute the labels for the testing set via Equation 8;

## Experiments

### Datasets

Following [2], we conduct experiments on three widely used image classification datasets, including **CUB-200-2011 (CUB)** [1], **SUN** [3], and **Animals with Attributes 2 (AWA2)** [2]. Detailed statistics of these datasets are summarized in Table 1.

- **CUB:** The Caltech-UCSD Birds-200-2011 (CUB) dataset is a fine-grained image classification dataset containing a total of 11,788 images from 200 bird species. Among these, 150 classes are used as seen classes, while the remaining 50 classes are treated as unseen classes. Each class is associated with 312 attributes, primarily describing color and shape characteristics of various bird parts. The significant attribute overlap across different classes in CUB poses a considerable challenge for zero-shot image classification models.
- **SUN:** The SUN dataset is a fine-grained scene image classification dataset containing 14,340 images from 717 different scenes. It consists of 645 seen classes and 72 unseen classes, with each class characterized by 102 attributes that describe geographic features, climatic characteristics, spatial structures, and other environmental factors. The diversity and subtle differences between scenes make accurate classification particularly challenging.
- **AWA2:** The Animals with Attributes 2 (AWA2) dataset is a coarse-grained image classification dataset comprising 37,332 images from 50 animal categories. It includes 40 seen classes and 10 unseen classes, with each class having

**Table 1. Statistic of three datasets.**

| Dataset | Detail | Attributes | Classes (S/U) | Total Images | Images (train) | Images (test S/U) |
|---------|--------|------------|---------------|--------------|----------------|-------------------|
| CUB | Fine-Grained | 312 | 150/50 | 11788 | 7057 | 1764/2967 |
| SUN | Fine-Grained | 102 | 645/72 | 14340 | 10320 | 2580/1440 |
| AWA2 | Coarse-Grained | 85 | 40/10 | 37322 | 23527 | 5882/7913 |

85 attributes that describe body parts, dietary characteristics, colors, and other distinguishing features. The wide variety of animal appearances and attributes makes robust image classification a significant challenge.

## Metrics

To comprehensively evaluate the classification performance of our proposed method, we adopt standard metrics used in zero-shot image classification.

For **Conventional Zero-Shot Learning (CZSL)**, where the test set consists solely of unseen classes, we use the **average per-class Top-1 accuracy** (*Acc*) as the primary evaluation metric. This metric reflects how well the model can accurately classify images belonging to unseen classes.

For **Generalized Zero-Shot Learning (GZSL)**, where the test set contains both seen and unseen classes, the evaluation metrics are more nuanced. We separately compute the **average precision** of seen classes (*S*) and unseen classes (*U*), as the inclusion of seen classes introduces a classification bias.

To quantify the model's overall classification capability across both seen and unseen classes, we calculate the **harmonic mean** (*H*), defined as $H = \frac{2 \cdot S \cdot U}{S + U}$. The harmonic mean effectively balances the performance between seen and unseen classes, providing a comprehensive assessment of the model's zero-shot classification ability.

## Implementation details

The CRAE model we propose is capable of end-to-end training. The backbone of our CRAE uses ResNet101 [22] that has been pre-trained on ImageNet-1K [23]. The input image resolution is $448 \times 448$, and the channel dimensions for class-level features, class prototypes, attribute prototypes, and attribute-level features are all 2048. Both the class prototype generator and the attribute prototype generator consist of 3 fully connected neural network layers. Each hidden layer contains 1024 units, and the output layer contains 2048 units. Relu activation functions are used in all fully connected layers. In attribute-level contrastive learning, *AF* is processed by a linear layer with an output dimension of 1024. SGD optimizer is selected to optimize the model, with a momentum parameter set to 0.9 and a weight decay rate set to $10^{-5}$. The initial learning rate is set to $10^{-3}$, with a learning rate decay applied every 10 epochs by a factor of 0.5. Regarding the loss functions, the temperature coefficients $\tau$ for attribute-level contrastive loss and class-level contrastive loss are set to 0.3 and 0.1 respectively. The positive sample selection ratio $\mu$ in attribute-level contrastive loss is set to 0.3, and the negative sample selection ratio $\varepsilon$ is set to 0.5. The scaling factor $\alpha$ in the classification loss is fixed at 25. The settings for loss coefficients on various datasets are presented in Table 2. CRAE adopts an episode-based training approach, where each batch of images contains *M* categories, and each category has *N* images. For the CUB and AWA2 datasets, $M = 16$ and $N = 2$, while for the SUN dataset, $M = 8$ and $N = 2$. During the model testing phase, $\gamma$ is set to 0.7 for the CUB and SUN datasets, and $\gamma$ is set to 1.0 for the AWA2 dataset.

## Comparison with state of the arts

To demonstrate the superiority of the proposed algorithm, we compare the performance of CRAE and the most recent state-of-the-art algorithms. These comparison algorithms are divided into two major categories: embedding-based zero-shot learning and generative-based zero-shot learning. Among them, there are 7 generative-based zero-shot learning algorithms: f-CLSWGAN [9], f-VAEGAN-D2 [10], LisGAN [11], TF-VAEGAN [14], Composer [12], HSVA [15], ICCE [13],

**Table 2. The loss coefficients setting for CRAE on each dataset.**

| Dataset | $\lambda_1$ | $\lambda_2$ | $\lambda_3$ | $\lambda_4$ |
|---------|-------------|-------------|-------------|-------------|
| CUB | 1.0 | 0.1 | 0.8 | $10^{-5}$ |
| SUN | 1.0 | 0.2 | 1.0 | $10^{-4}$ |
| AWA2 | 1.0 | 1.0 | 1.0 | $10^{-6}$ |

and 10 embedding-based zero-shot learning algorithms: TCN [24], AREN [25], DAZLE [7], RGEN [26], APN [8], DCEN [27], DPPN [28], GEM-ZSL [29], MSDN [30], EMP [21].

**Performance on CZSL.** The CRAE model is an embedding-based zero-shot learning method capable of performing both conventional zero-shot learning tasks and generalized zero-shot learning tasks. In this section, experiments are conducted on three commonly used zero-shot learning datasets, and compared with 7 generative-based zero-shot learning methods and 10 embedding-based zero-shot learning methods. Table 3 presents the corresponding experimental results. For the CZSL setting, the proposed CRAE model achieves the best results on all three datasets, with classification accuracies of 79.4%, 67.7%, and 75.8%, respectively. For the coarse-grained dataset AWA2, the CRAE model outperforms the second-best algorithm by 2.2%, indicating its ability to extract discriminative features from images effectively and transfer knowledge from seen classes to unseen classes. For the fine-grained datasets CUB and SUN, the CRAE model surpasses the second-best algorithm by 1.6% and 1.7%, respectively, demonstrating its capability to differentiate semantically similar categories and exhibit superior generalization performance.

**Performance on GZSL.** Table 3 also gives the results of our method and other state-of-the-art methods in the GZSL setting. In the GZSL setting, $H$ is an important metric in generalized zero-shot learning, as it comprehensively reflects the performance of the model on both unseen and seen classes. As shown in Table 3, the CRAE model achieves the highest $H$ on all three datasets, with results of 74.6% on the CUB dataset, 45.0% on the SUN dataset, and 77.2% on the AWA2 dataset. For the challenging SUN dataset, the proposed model outperforms the second-best algorithm by 1.7% on $H$ and 0.5% on $S$, indicating the model's ability to learn semantic attribute information through attribute embedding, reducing the gap between visual and semantic features, and thus improving algorithm performance. The $U$ and $S$ gaps of the DAZLE model are large, at 28%, while the gaps between these two metrics for the CRAE model are less than 10%, demonstrating the model's effective balancing of classification accuracy between seen and unseen classes. For the fine-grained CUB dataset, the CRAE model outperforms the second-best algorithm by 0.2% on $H$ and 0.6% on $U$, indicating the model's ability to enhance representation discriminability through class-level contrastive learning, reducing confusion between visually

**Table 3. Results (%) of CRAE and other state-of-the-art methods on CUB, SUN and AWA2. The best results are in bold. The second-best results are in _italic_. Symbol "–" denotes no results are recorded.**

| Type | Methods | CUB | | | | SUN | | | | AWA2 | | | |
|---|---|---|---|---|---|---|---|---|---|---|---|---|---|
| | | CZSL | GZSL | | | CZSL | GZSL | | | CZSL | GZSL | | |
| | | _Acc_ | _U_ | _S_ | _H_ | _Acc_ | _U_ | _S_ | _H_ | _Acc_ | _U_ | _S_ | _H_ |
| Generative | f-CLSWGAN [9] | 57.3 | 43.7 | 57.7 | 49.7 | 60.8 | 42.6 | 36.6 | 39.4 | 68.2 | 57.9 | 61.4 | 59.6 |
| | f-VAEGAN-D2 [10] | 61.0 | 48.4 | 60.1 | 53.6 | 64.7 | 45.1 | 38.0 | 41.3 | 71.1 | 57.6 | 70.6 | 63.5 |
| | LisGAN [11] | 58.8 | 46.5 | 57.9 | 51.6 | 61.7 | 42.9 | 37.8 | 40.2 | - | - | - | - |
| | TF-VAEGAN [14] | 64.9 | 52.8 | 64.7 | 58.1 | _66.0_ | 45.6 | _40.7_ | 43.0 | 72.2 | 59.8 | 75.1 | 66.6 |
| | Composer [12] | 69.4 | 56.4 | 63.8 | 59.9 | 62.6 | **55.1** | 22.0 | 31.4 | 71.5 | 62.1 | 77.3 | 68.8 |
| | HSVA [15] | - | 52.7 | 58.3 | 55.3 | - | 48.6 | 39.0 | _43.3_ | - | 56.7 | 79.8 | 66.3 |
| | ICCE [13] | - | 67.3 | 65.5 | 66.4 | - | - | - | - | - | 65.3 | 82.3 | 72.8 |
| Embedding | TCN [24] | 59.5 | 52.6 | 52.0 | 52.3 | 61.5 | 31.2 | 37.3 | 34.0 | 71.2 | 61.2 | 65.8 | 63.4 |
| | AREN [25] | 71.8 | 38.9 | 78.7 | 52.1 | 60.6 | 19.0 | 38.8 | 25.5 | 67.9 | 15.6 | **92.9** | 26.7 |
| | DAZLE [7] | 66.0 | 56.7 | 59.6 | 58.1 | 59.4 | _52.3_ | 24.3 | 33.2 | 67.9 | 60.3 | 75.7 | 67.1 |
| | RGEN [26] | 76.1 | 60.0 | 73.5 | 66.1 | 63.8 | 44.0 | 31.7 | 36.8 | _73.6_ | _67.1_ | 76.5 | 71.5 |
| | APN [8] | 72.0 | 65.3 | 69.3 | 67.2 | 61.6 | 41.9 | 34.0 | 37.6 | 68.4 | 57.1 | 72.4 | 63.9 |
| | DCEN [27] | - | 63.8 | 78.4 | 70.4 | - | 43.7 | 39.8 | 41.7 | - | 62.4 | 81.7 | 70.8 |
| | DPPN [28] | - | 70.2 | 77.1 | 73.5 | - | 47.9 | 35.8 | 41.0 | - | 63.1 | 86.8 | _73.1_ |
| | GEM-ZSL [29] | _77.8_ | 64.8 | 77.1 | 70.4 | 62.8 | 38.1 | 35.7 | 36.9 | 67.3 | 64.8 | 77.5 | 70.6 |
| | MSDN [30] | 76.1 | 68.7 | 67.5 | 68.1 | 65.8 | 52.2 | 34.2 | 41.3 | 70.1 | 62.0 | 74.5 | 67.7 |
| | _PFRN_ [31] | 77.1 | 72.7 | 75.0 | 73.8 | _66.3_ | 55.5 | 32.3 | 40.9 | 71.3 | 68.6 | 84.3 | 75.6 |
| | _VENet-Res_ [32] | 78.1 | 71.5 | 73.0 | 72.2 | 66.7 | 49.9 | 36.4 | 42.1 | 73.8 | 67.5 | 79.8 | 73.2 |
| | EMP [21] | - | _70.8_ | **78.4** | 74.4 | - | 47.4 | 36.0 | 40.9 | - | 62.1 | 87.5 | 72.7 |
| | **CRAE(Ours)** | **79.4** | **71.4** | _78.2_ | **74.6** | **67.7** | 49.6 | **41.2** | **45.0** | **75.8** | **68.6** | _88.1_ | **77.2** |

similar categories during classification. Additionally, the seen class accuracy of the AREN and GEM-ZSL models is significantly higher than their unseen class accuracy, indicating bias towards seen classes. The proposed model alleviates bias issues and improves the classification performance of unseen classes. For the coarse-grained AWA2 dataset, where the visual differences between animals with the same attribute are significant, the CRAE model outperforms the second-best algorithm by 4.1% on $H$ and 1.5% on $U$, indicating significant improvement and effective mitigation of domain shift problems, making it more suitable for image classification tasks. Most existing methods in generalized zero-shot learning tasks only achieve high results on either seen or unseen classes. However, the proposed model achieves a good balance between the two metrics because it combines class representation learning and attribute embedding learning, enhancing the discriminability of global visual features and attribute-level features. Overall, the proposed CRAE model achieves good results in both conventional zero-shot learning tasks and generalized zero-shot learning tasks.

### Ablation studies

**Component analysis.** In this section, we conducted the ablation experiments on the CUB, SUN, and AWA2 datasets to analyze the effectiveness of different modules in CRAE for generalized zero-shot learning. The results of the ablation experiments on the three datasets are shown in Table 4, where the methods with the highest classification accuracy are marked in red. Extracting discriminative attribute-level features related to attributes is crucial for zero-shot learning to overcome the domain shift problem and enhance visual-semantic correlations. The baseline model utilized only the classification loss in class representation learning and the regression loss in attribute embedding learning, denoted as $\mathcal{L}_{cls} + \lambda_1 \mathcal{L}_{reg}$, similar to the APN model. The difference lies in the baseline model completing classification in the visual embedding space using cosine similarity-based cross-entropy loss, while the APN model performs classification in the semantic embedding space using dot product similarity-based cross-entropy loss. Then, we successively added attribute alignment loss and attribute-level contrastive loss without hard sample selection as the second and third sub-models on the basis of the baseline model. To validate the importance of hard sample selection, we added attribute-level contrastive loss based on hard sample selection as the fourth sub-model on the basis of the second sub-model. Subsequently, we added attribute-based adaptive softmax activation function and class-level contrastive loss as the fifth sub-model and the final CRAE model on the basis of the fourth sub-model.

As shown in Table 4, each additional loss or mechanism demonstrated its performance improvement, thereby continuously enhancing the model's classification effectiveness. After adding attribute alignment loss, CUB and SUN improved by 0.9% and 1.4%, respectively, on $H$, while AWA2 dataset showed a slight decrease, indicating the effectiveness of generating attribute prototypes and using attribute alignment loss. When adding regular attribute layer contrastive loss, the model improved by 0.1%, 0.4%, and 0.8% on $H$ for the three datasets, respectively, demonstrating that attribute-level contrastive learning can reduce the differentiation of the same attribute among different visual targets in the embedding space, but the discriminative nature of attribute embedding space still needs improvement. The selection of positive and negative samples has a significant impact on contrastive learning. When introducing the hard sample selection mechanism in the

**Table 4. Results (%) of ablation study on three datasets. The best results are marked in bold. HS means hard samples selection.**

| Methods | CUB | | | SUN | | | AWA2 | | |
|---|---|---|---|---|---|---|---|---|---|
| | U | S | H | U | S | H | U | S | H |
| $\mathcal{L}_{cls} + \lambda_1 \mathcal{L}_{reg}$ | 63.7 | 77.9 | 70.1 | 40.9 | 41.9 | 41.4 | 63.1 | 87.9 | 73.5 |
| $+\lambda_2 \mathcal{L}_{aal}$ | 67.8 | 74.5 | 71.0 | 43.6 | 42.0 | 42.8 | 62.8 | 88.4 | 73.4 |
| $+\lambda_3 \mathcal{L}_{acl}$ w/o HS | 65.5 | 77.8 | 71.1 | 47.2 | 39.8 | 43.2 | 65.4 | 85.9 | 74.2 |
| $+\lambda_3 \mathcal{L}_{acl}$ w HS | 69.8 | 78.1 | 73.7 | 43.8 | **42.8** | 43.3 | 67.5 | 86.5 | 75.8 |
| +attribute-based adaptive softmax | 70.5 | **78.6** | 74.3 | 48.2 | 41.2 | 44.5 | 68.1 | **88.6** | 77.0 |
| $+\lambda_4 \mathcal{L}_{ccl}$ | **71.4** | 78.2 | **74.6** | **49.6** | 41.2 | **45.0** | **68.6** | 88.1 | **77.2** |

attribute-level contrastive loss, the model improved by 2.6%, 0.1%, and 1.6% on $H$ for the three datasets, respectively, indicating that the hard sample selection mechanism can significantly enhance the discriminative nature of the attribute embedding space. At this point, the attribute-level features may contain more noise from irrelevant local features. When adding attribute-based adaptive softmax activation function, $H$ improved by 0.6%, 1.2%, and 1.2% on the three datasets, respectively, indicating that the attribute-based adaptive softmax activation function can reduce noise in attribute-level features and thereby improve model performance. After adding class-level contrastive loss, $H$ increased by 0.3%, 0.5%, and 0.2% on CUB, SUN, and AWA2, respectively, indicating that class-level contrastive loss can enhance the discriminative nature of global visual features in the visual embedding space, enabling the model to differentiate visually similar but different categories.

**Training method analysis.** This chapter adopts the episode-based training approach to enhance the generalization performance of the model. In this training setup, each batch randomly samples $M$ classes, with each class containing $N$ images. Another commonly used training method is based on random sampling, where a certain number of images are randomly sampled from the entire training set. A comparison of the experimental results of different training methods is presented in Table 5, with the optimal results highlighted in red font. For the episode-based training method, $M$ is set to [4,8,16], while $N$ is fixed at 2. For the random sampling training method, there are no $M$ and $N$, and the number of images per batch is set to 32. From the table, it can be observed that generally, the episode-based training method yields higher results in generalized zero-shot learning experiments compared to the random sampling training method. This is because the episode-based training method ensures a consistent number of classes, allowing for more efficient utilization of limited samples, which stabilizes the number of positive and negative samples in attribute-level and class-level contrastive learning, thus enhancing the effectiveness of contrastive learning. On the other hand, the random sampling training method fails to ensure the stability of positive and negative samples in attribute-level and class-level contrastive learning. Moreover, it may even result in batches without any positive samples, thereby diminishing the intended effect of contrastive learning. Additionally, on the CUB dataset, when $M = 16$ and $N = 2$, the model achieves the highest $H$, surpassing the next best training method by 0.7%; on the SUN dataset, when $M = 8$ and $N = 2$, the model's $H$ exceeds the next best training method by 1.3%; and on the AWA2 dataset, when $M = 16$ and $N = 2$, the model's $H$ surpasses the next best training method by 2.6%. However, on the SUN dataset, excessively large $M$ values may lead to overfitting of the training data. Therefore, it is necessary to choose a suitable training method based on the dataset and hardware facilities available.

## Hyperparameter analysis

**The importance of hard samples selection.** Correctly selecting positive and negative samples is crucial for achieving high-performance contrastive learning. For a given anchor sample, defining same-class samples with lower similarity

**Table 5. Influence of training method on GZSL results (%). The best results are marked in bold.**

| CUB | Training Methods | M-way | N-shot | U | S | H |
|-----|------------------|-------|--------|------|--------|--------|
|  | random sampling | - | - | 69.2 | 76.9 | 72.9 |
|  | episode-based | 4 | 2 | 63.8 | 73.1 | 68.1 |
|  |  | 8 | 2 | 69.7 | **78.7** | 73.9 |
|  |  | 16 | 2 | **71.4** | 78.2 | **74.6** |
| SUN | random sampling | - | - | 49.4 | 39.1 | 43.7 |
|  | episode-based | 4 | 2 | 46.5 | 30.5 | 36.9 |
|  |  | 8 | 2 | **49.6** | 41.2 | **45.0** |
|  |  | 16 | 2 | 42.2 | **44.3** | 43.2 |
| AWA2 | random sampling | - | - | 62.9 | 86.2 | 72.7 |
|  | episode-based | 4 | 2 | 60.9 | 88.0 | 71.9 |
|  |  | 8 | 2 | 63.5 | **90.3** | 74.6 |
|  |  | 16 | 2 | **68.6** | 88.1 | **77.2** |

as hard positive samples, and different-class samples with higher similarity as hard negative samples, is essential. Simple positive and negative samples make it easier for contrastive learning to learn ambiguous boundary information, while hard positive and negative samples enable contrastive learning to learn clearer boundary information. This subsection analyzes the effects of the positive sample selection ratio $\mu$ and the negative sample selection ratio $\varepsilon$ on conventional zero-shot learning and generalized zero-shot learning, reporting the experimental results of these two parameters on the CUB and SUN datasets, as shown in Fig 2, where *Acc* represents the indicator for conventional zero-shot learning, and *H* represents the indicator for generalized zero-shot learning. Specifically, $\mu$ and $\varepsilon$ vary between [0.1,0.2,0.3,0.4,0.5] and the remaining hyperparameters in the model are fixed. From the Fig 2, it can be observed that for the CUB and SUN datasets, the model achieves optimal performance when $\mu = 0.3$ and $\varepsilon = 0.5$. For the CUB dataset, a noticeable decrease in model performance is observed at $\mu = 0.5$ and $\varepsilon = 0.5$, indicating that removing too many simple positive and negative samples may excessively reduce the number of training samples, affecting the model's generalization performance. Under the setting of $\mu = 0.1$ and $\varepsilon = 0.1$, optimal results are not achieved on both datasets, indicating that removing too few positive and negative samples also cannot yield the optimal positive and negative sample sets. Reasonable selection of the ratio for selecting hard samples helps better constrain the attribute-level features with the contrastive loss, reduce the differences

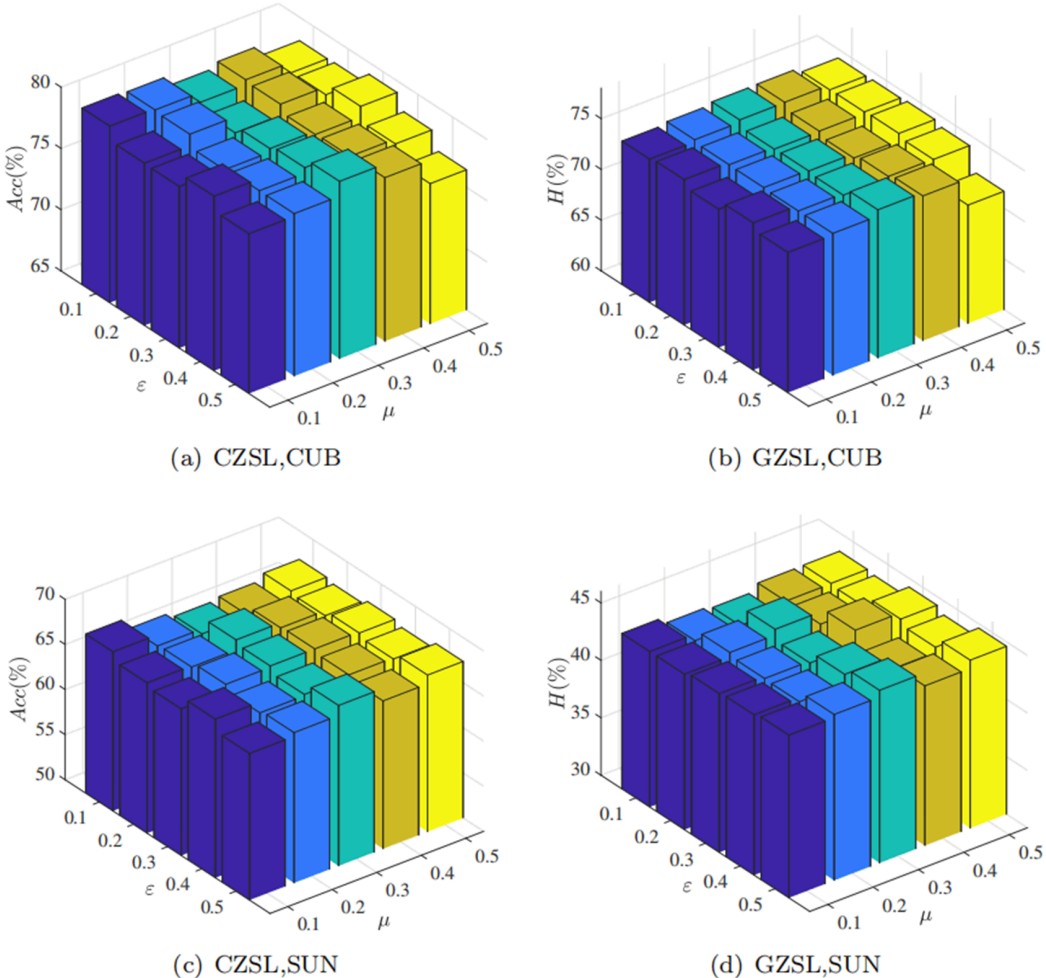

(a) CZSL,CUB  (b) GZSL,CUB

(c) CZSL,SUN  (d) GZSL,SUN

**Fig 2. The classification results on CUB and SUN with different values of $\mu$ and $\varepsilon$.**

in the embedding space of the same attribute on different visual targets, and thus learn discriminative attribute embedding space.

While the empirically selected thresholds demonstrate good generalization across datasets, fixed thresholds may limit adaptability in more complex scenarios. Therefore, future work may focus on developing adaptive thresholding strategies, such as leveraging percentile-based statistics on similarity distributions or integrating learnable modules that dynamically adjust $\mu$ and $\varepsilon$ according to training dynamics. Such approaches could enhance the robustness and generalization capability of the model in diverse zero-shot learning tasks.

**Evaluation of $\lambda_3$.** We evaluate the effect of $\lambda_3$ for our model on CUB and SUN datasets in GZSL. $\lambda_3$ controls the model's attention to the contrastive loss at the attribute level. Fig 3 illustrates the impact of $\lambda_3$ on the CUB and SUN datasets, with evaluation metrics being $U$, $S$, and $H$ in generalized zero-shot learning. Specifically, $\lambda_3$ varies between 0.1 and 1, with other hyperparameters fixed. For the CUB dataset, the model achieves optimal results on $H$ when $\lambda_3 = 0.8$, on $U$ when $\lambda_3 = 0.9$, and the highest $S$ when $\lambda_3 = 0.2$. On the SUN dataset, the model achieves optimal $H$ when $\lambda_3 = 1.0$, the highest $U$ when $\lambda_3 = 0.8$, and the highest $S$ when $\lambda_3 = 0.3$. Further observation of Fig 3 reveals that as $\lambda_3$ increases, $H$ shows an upward trend for both datasets, indicating that the coefficient of the contrastive loss based on difficult sample selection at the attribute level is a key parameter for enhancing model performance. Moreover, after $\lambda_3 > 0.5$, the variation in $H$ on both datasets is small, indicating that the CRAE model exhibits good robustness.

**Evaluation of $\lambda_4$.** We evaluate the effect of $\lambda_4$ for our model on CUB and SUN datasets in GZSL. $\lambda_4$ controls the model's attention to the contrastive loss at the class level. Fig 4 illustrates the impact of $\lambda_4$ on the CUB and SUN datasets, with evaluation metrics being $U$, $S$, and $H$ in generalized zero-shot learning. Specifically, $\lambda_4$ is restricted to the interval $[10^{-8}, 10^{-7}, 10^{-6}, 10^{-5}, 10^{-4}, 10^{-3}]$, with other hyperparameters fixed. It can be observed that $\lambda_4$ has a minor impact on model performance, indicating that the model is not sensitive to $\lambda_4$. For the CUB dataset, the optimal parameter values for $U$, $S$, and $H$ are $10^{-6}$, $10^{-3}$, and $10^{-5}$, respectively. For the SUN dataset, the optimal parameter values for $U$, $S$, and $H$ are $10^{-2}$, $10^{-7}$, and $10^{-4}$, respectively. $\lambda_4$ being too small or too large does not lead to optimal model performance, hence it should be set to a reasonable value within the specified interval.

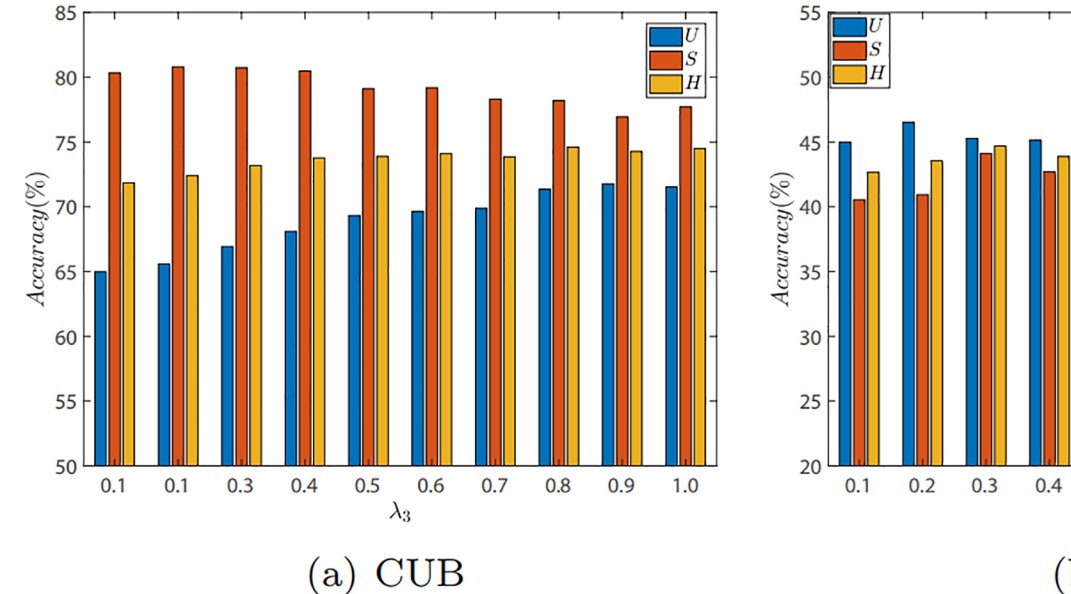

(a) CUB  (b) SUN

**Fig 3**. The GZSL results on CUB and SUN with different values of $\lambda_3$.

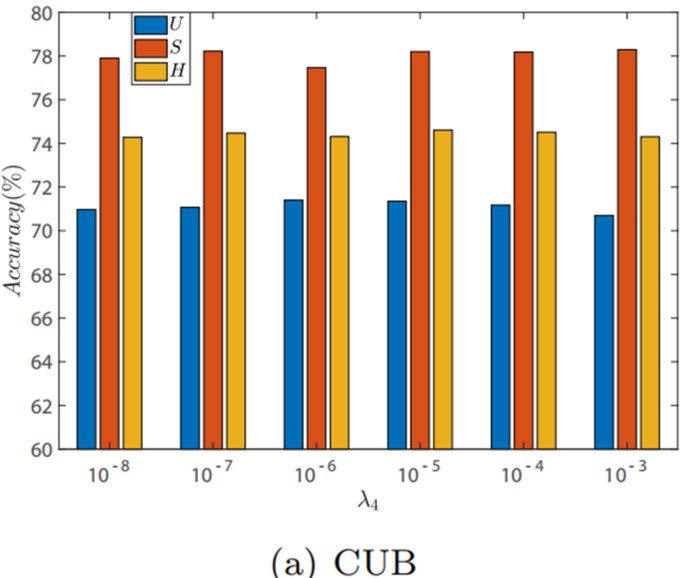

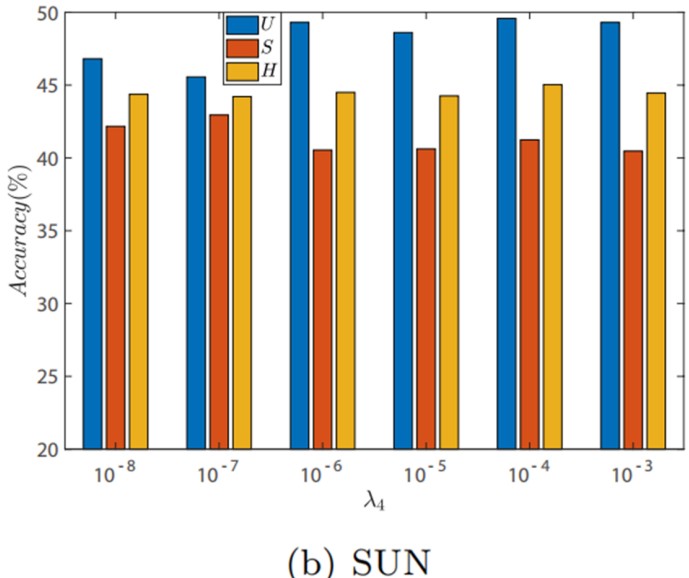

(a) CUB　　　　　　　　　　　　　　　(b) SUN

**Fig 4. The GZSL results on CUB and SUN with different values of $\lambda_4$.**

## Qualitative results

**t-SNE visualizations of class representations.** As shown in Fig 5, t-SNE visualization [33] is employed to illustrate the class representations (global visual features $g(x)$) of seen and unseen classes on the test datasets of CUB and AWA2. For experimentation, 10 seen classes and 10 unseen classes are randomly selected from the CUB dataset, along with the top 10 seen classes and all unseen classes from the AWA2 dataset. Different colors represent different classes. The visualization results reveal that, for both datasets, the global visual features of the same class tend to cluster together, indicating good intra-class consistency, while the features of different classes are dispersed, indicating distinct inter-class differences. This demonstrates that the CRAE model can learn highly discriminative visual embedding spaces, which is advantageous for image classification tasks, as the class-level contrastive learning constrains the global visual features. Additionally, the visualization results of unseen classes on both datasets demonstrate the model's proficiency in distinguishing unseen class images. This is attributed to the attribute embedding learning branch assisting the class representation learning branch, enriching the global visual features with deeper semantic information.

**t-SNE visualizations of attribute features.** To assess the discriminative nature of attribute embedding space, 15 attributes were randomly selected from each of the three datasets, resulting in a total of 2048 attribute-level features. These features were then visualized using t-SNE, as shown in Fig 6. Features of the same color share the same attribute label, while features of different colors have different attribute labels. It can be observed that the attribute features of different attributes are easily separated in the attribute embedding space, with clear boundaries between them. Conversely, features of the same attribute tend to cluster together, indicating good intra-class consistency. This confirms that the CRAE model can reduce the differences between the same attribute in the embedding space and has the ability to accurately identify attributes, which helps overcome domain shift issues. This is primarily because the attribute-level contrastive loss based on hard sample selection can constrain the distribution of attribute-level features in the attribute embedding space, enhancing the discriminative nature of attribute-level features. Further observation of Fig 6 reveals that some attributes (e.g., mouth attribute represented by red dots in Fig 6(3) ) have more feature points, while some attributes (e.g., hoof attribute represented by green dots in Fig 6(3)) have fewer feature points, reflecting the different frequencies

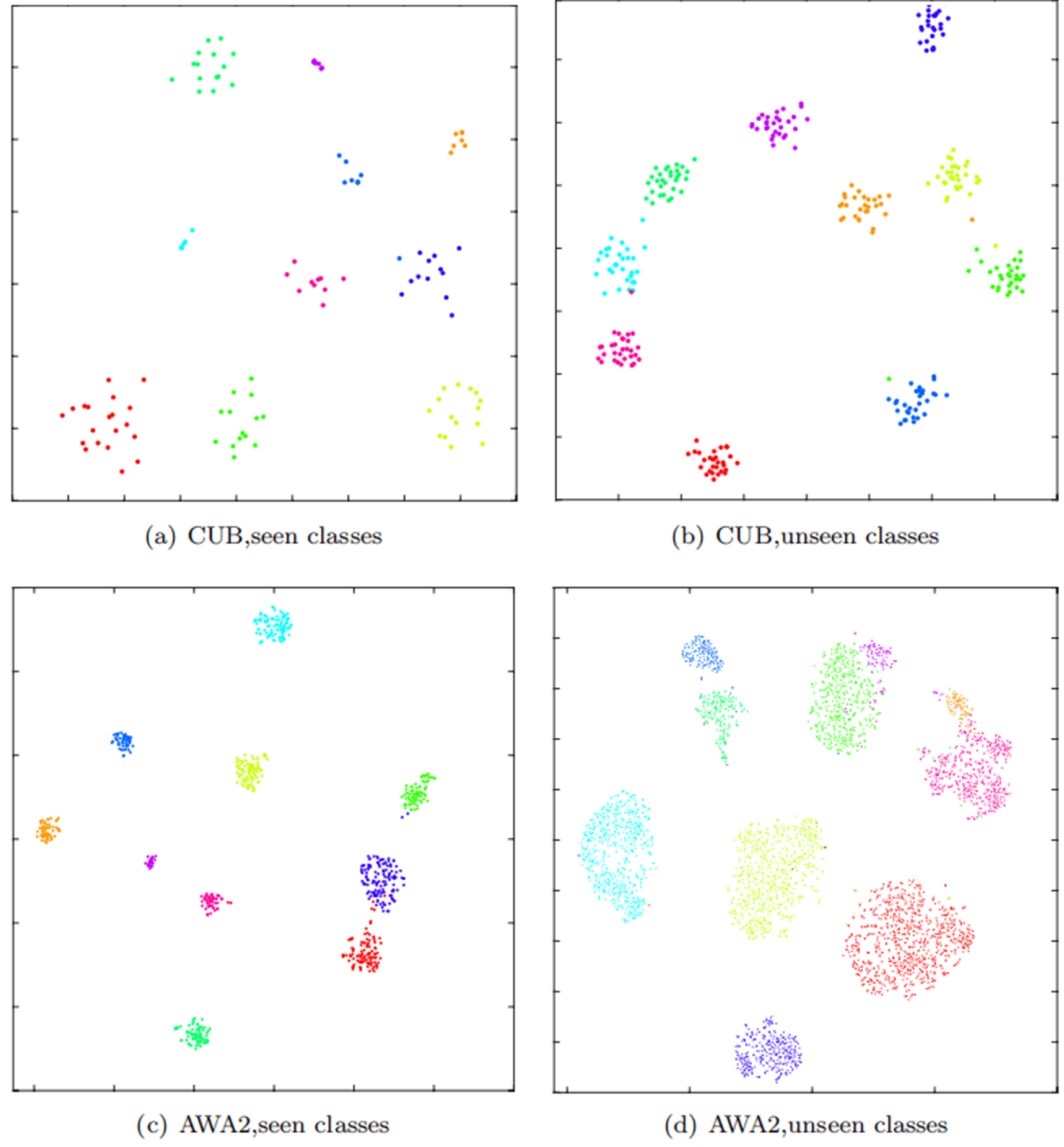

(a) CUB,seen classes  (b) CUB,unseen classes

(c) AWA2,seen classes  (d) AWA2,unseen classes

**Fig 5**. The t-SNE visualization of class representations for seen and unseen classes on CUB and AWA2.

of occurrence of different attributes in the dataset. Moreover, Fig 6 also reflects the semantic similarity between different attributes; for example, the semantic similarity between tail and hoof is less than that between claw and hoof.

**Attribute localization.** To evaluate the effectiveness of attribute localization in the proposed model, visualizations of the normalized attribute feature maps $A(x)$ in the CRAE model are presented on the CUB and SUN datasets, as shown in Fig 7. Specifically, two test images were selected from each dataset, and these images were input into the trained model. From the output of the attribute filtering network, attribute feature maps for each attribute were obtained. Several attribute feature maps were chosen, and corresponding heatmaps were generated using the Grad-CAM algorithm [34]. Darker areas in the heatmap indicate regions of the image that the model pays more attention to regarding a specific attribute, reflecting the association between attributes and local visual features. If the model's attribute localization is accurate, then the attribute-level features obtained from the attribute embedding module will more accurately represent the attribute.

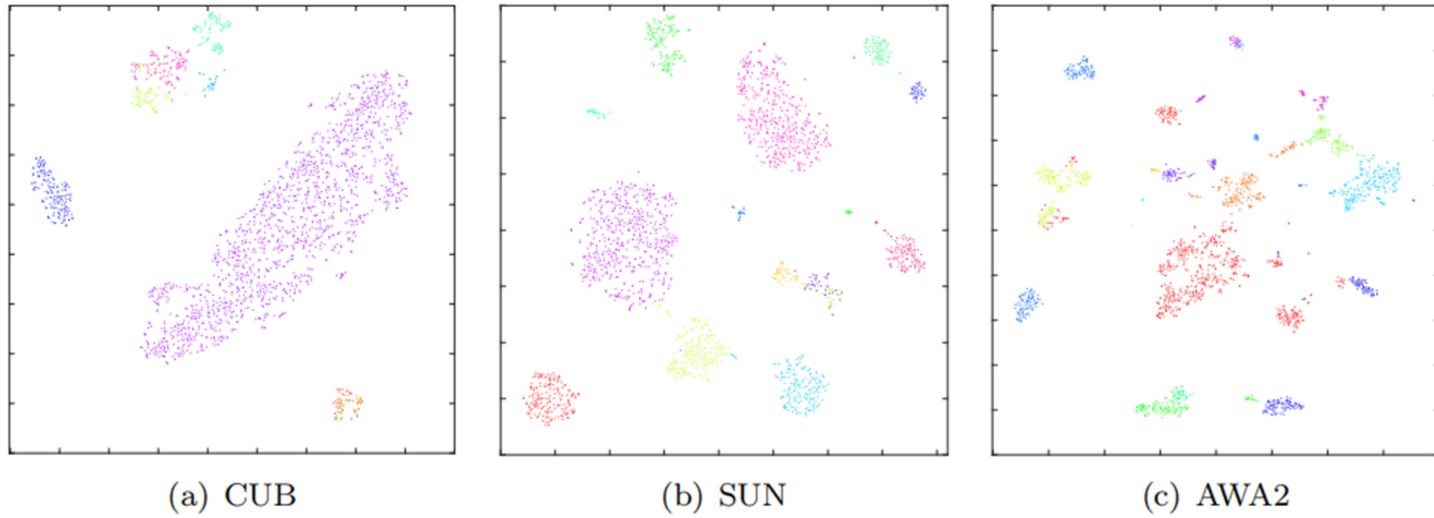

**Fig 6**. **The t-SNE visualization of attribute features on CUB, SUN, and AWA2.**

Conversely, attribute-level features may be mixed with irrelevant local visual features, thereby weakening the effectiveness of attribute embedding learning.

Fig 7(1) demonstrates that the CRAE model accurately localizes the attributes on the CUB dataset, particularly when localizing head and foot attributes. This could be because head and foot attributes mostly appear in the images, while other attributes may not appear due to issues such as shooting angle or occlusion. Fig 7(2) illustrates that the CRAE model can still localize key attributes on the challenging SUN dataset, such as trees, clouds, and oceans. However, localization of abstract attributes is more ambiguous, such as the action attribute "vacationing", because abstract attributes typically do not directly appear in the image and require inference from various parts of the image.

## Convergence analysis

The CRAE model combines class representation learning and attribute embedding learning branches during training, making it an end-to-end model. Therefore, this section analyzes the stability of the model during training, as shown in Fig **??**. In the experiments, the maximum number of training iterations was set to 15. The changes in the loss function value and the generalized zero-shot learning metric $H$ with the number of iterations were recorded on both the CUB and SUN datasets. The red curve represents the variation of $H$, while the blue curve represents the variation of the loss function value. It can be observed from the figure that in the first 10 rounds, there is a significant decrease in the loss function and a noticeable increase on $H$, indicating the rapid convergence of the model. Additionally, in the subsequent 3 rounds, there is no significant change on $H$, suggesting the stability of the model during training. It is evident from the graph that the training on the CUB dataset is more stable compared to that on the SUN dataset. This is attributed to the fact that the CUB dataset contains 200 categories, whereas the SUN dataset contains 717 categories, indicating that training becomes more challenging as the number of categories increases in image classification tasks.

## Conclusion

In this work, we address critical limitations in current zero-shot image classification approaches, particularly their inability to handle significant visual variations within the same attribute and the pervasive noise introduced by irrelevant local

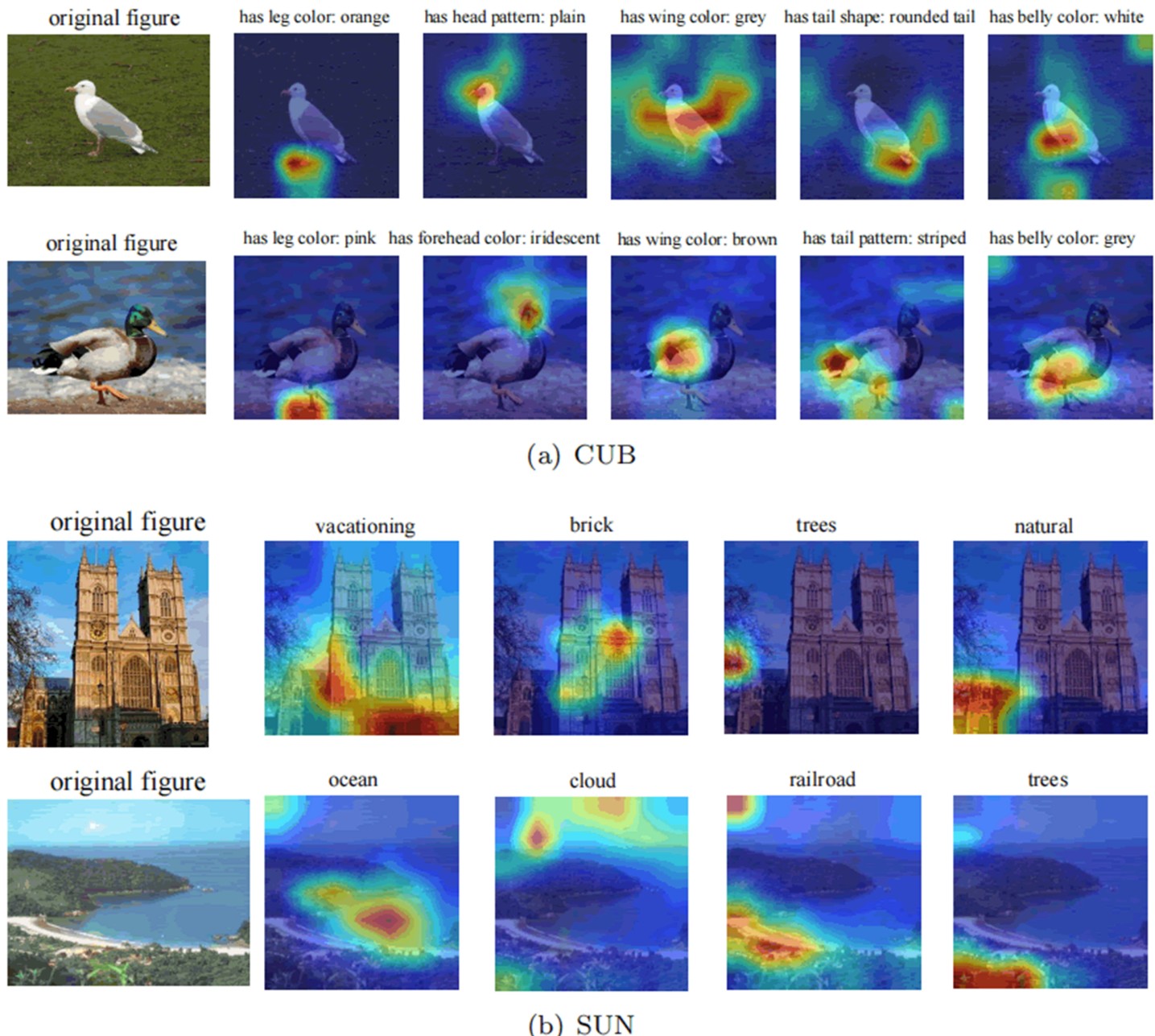

**Fig 7**. Visualization results of attribute feature maps on CUB and SUN datasets.

features. To overcome these challenges, we propose the CRAE algorithm, a novel zero-shot image classification framework that effectively combines class representation learning with attribute embedding learning.

The core idea of CRAE is to jointly optimize class-level and attribute-level feature representations to enhance classification accuracy. In attribute embedding learning, we leverage attribute-level contrastive learning with hard sample selection to constrain the distribution of attribute-level features, reducing discrepancies of the same attribute across different

visual targets. Additionally, we introduce an adaptive softmax activation function that dynamically normalizes attribute feature maps, significantly mitigating noise and improving attribute discriminability.

In class representation learning, we incorporate class-level contrastive learning to strengthen the discriminative power of global visual features, enabling the model to better distinguish between visually similar categories. By seamlessly integrating these two components, CRAE achieves a robust and accurate classification performance in zero-shot scenarios.

Comprehensive experiments conducted on three widely-used benchmark datasets, including CUB, SUN, and AWA2, demonstrate the superior performance of CRAE compared to state-of-the-art methods. These results confirm the effectiveness of our approach in tackling the complex challenges of zero-shot image classification and establish CRAE as a promising solution for real-world applications.

Despite the strong performance of CRAE, several challenges remain for future exploration. First, the model's performance may be affected by missing or inaccurate attribute annotations, which can lead to misaligned feature representations. Second, the presence of noisy labels in training data, especially for fine-grained categories, may degrade the robustness of contrastive learning. Third, class imbalance between seen and unseen categories remains an inherent challenge in zero-shot settings. Future research may explore robust attribute completion techniques, noise-resilient learning strategies, and meta-learning-based adaptation to further improve the generalization and applicability of zero-shot learning methods.

## Author contributions

**Conceptualization:** Huabo Shen.

**Data curation:** Xiaodong Sun.

**Formal analysis:** Xiaodong Sun.

**Funding acquisition:** Xiaodong Sun.

**Project administration:** Qin Li, Youmin Hu.

**Resources:** Changgeng Li.

**Software:** Changgeng Li.

**Supervision:** Youmin Hu, Qinmiao Zhu.

**Validation:** Qin Li, Qinmiao Zhu.

**Visualization:** Changgeng Li.

**Writing – original draft:** Huabo Shen.

**Writing – review & editing:** Qin Li, Youmin Hu, Qinmiao Zhu.

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
