## [Decision Letter · Decision Letter 0]

20 Jun 2025

PONE-D-25-22333Zero-Shot Image Classification Based on Class Representation Learning and Attribute Embedding LearningPLOS ONE

Dear Dr. Li,

Thank you for submitting your manuscript to PLOS ONE. After careful consideration, we feel that it has merit but does not fully meet PLOS ONE’s publication criteria as it currently stands. Therefore, we invite you to submit a revised version of the manuscript that addresses the points raised during the review process.

We look forward to receiving your revised manuscript.

Kind regards,

Haofeng Zhang

Academic Editor

PLOS ONE

Journal Requirements: 

 [This work is supported by Natural Science Foundation of Guangdong Province under Grant 2023A1515011845 and Shenzhen Science and Technology Major Special Project under Grant KJZD20231023092602004).]. 

[This work is supported by Natural Science Foundation of Guangdong Province under Grant 2023A1515011845 and Shenzhen Science and Technology Major Special Project under Grant KJZD20231023092602004).]

  [This work is supported by Natural Science Foundation of Guangdong Province under Grant 2023A1515011845 and Shenzhen Science and Technology Major Special Project under Grant KJZD20231023092602004).]. 

4. Thank you for uploading your study's underlying data set. Unfortunately, the repository you have noted in your Data Availability statement does not qualify as an acceptable data repository according to PLOS's standards.

5. Please amend the manuscript submission data (via Edit Submission) to include author Huabo Shen, Xiaodong Sun, Youmin Hu, Changgeng Li and Qinmiao Zhu.

Reviewers' comments:

Reviewer's Responses to Questions

**Comments to the Author**

1. Is the manuscript technically sound, and do the data support the conclusions?

Reviewer #1: Yes

Reviewer #2: Yes

2. Has the statistical analysis been performed appropriately and rigorously?

Reviewer #1: Yes

Reviewer #2: Yes

3. Have the authors made all data underlying the findings in their manuscript fully available?

Reviewer #1: Yes

Reviewer #2: Yes

4. Is the manuscript presented in an intelligible fashion and written in standard English?

Reviewer #1: Yes

Reviewer #2: Yes

5. Review Comments to the Author

Reviewer #1: Overall Evaluation:

This paper proposes a method named CRAE (Class Representation and Attribute Embedding) to address the limitation that existing approaches often fail to capture significant intra-attribute visual variations. The method introduces an attribute-based adaptive softmax activation to dynamically normalize attribute feature maps and further incorporates attribute-level contrastive learning and class representation learning with attribute embedding learning.

Comments:

1. Figure 1 is missing from the main manuscript, which makes it difficult to fully understand the method’s pipeline.

2. The attribute-level contrastive embedding bears strong similarity to prior work [1], but the paper does not clearly differentiate its approach or provide a comparative discussion.

3. Although the adaptive softmax activation is claimed as a key contribution, the paper lacks a dedicated section or sufficient analysis highlighting this component.

4. The baseline methods used for comparison are outdated. Including more recent state-of-the-art approaches would strengthen the experimental validation.

[1] Semantic Contrastive Embedding for Generalized Zero-Shot Learning. IJCV, 2022.

Reviewer #2: The manuscript introduces a well-designed and effective zero-shot learning method named CRAE, which jointly integrates class representation learning and attributes embedding learning to address challenges in conventional and generalized zero-shot image classification. The paper is well-structured and clearly written. The proposed approach is technically sound and demonstrates competitive performance on multiple datasets showing improvements over the comparable methods.

The following suggestions may further improve the quality and clarity of the paper:

1. While the combination of adaptive softmax and dual contrastive learning is compelling, a more in-depth theoretical explanation or deep analysis would be also necessary.

2. The choice of hard positive/negative thresholds (μ and ε) is important but appears empirically chosen. It should discuss the rationale and potential to generalize or learn these thresholds adaptively.

3. The ablation study would benefit from more visual evidence (e.g., t-SNE plots) comparing feature distributions.

4. May, it would be better to give a brief discussion on potential challenges (e.g., missing attributes, noisy labels, class imbalances) and future directions.

5. Some equations (e.g., Eq. 5) are complex and could be reformatted for readability.

6. PLOS authors have the option to publish the peer review history of their article (what does this mean?). If published, this will include your full peer review and any attached files.

Reviewer #1: No

Reviewer #2: No

---

## [Author Response · Author response to Decision Letter 1]

11 Aug 2025

The response to reviewers is attached as a specific document in the "attached files"

---

## [Decision Letter · Decision Letter 1]

5 Sep 2025

Zero-Shot Image Classification Based on Class Representation Learning and Attribute Embedding Learning

PONE-D-25-22333R1

Dear Dr. Li,

We’re pleased to inform you that your manuscript has been judged scientifically suitable for publication and will be formally accepted for publication once it meets all outstanding technical requirements.

Kind regards,

Haofeng Zhang

Academic Editor

PLOS ONE

Additional Editor Comments (optional):

Both reviewers suggested to accept this paper.

Reviewer #1:

Reviewer #2:

Reviewers' comments:

Reviewer's Responses to Questions

**Comments to the Author**

1. If the authors have adequately addressed your comments raised in a previous round of review and you feel that this manuscript is now acceptable for publication, you may indicate that here to bypass the “Comments to the Author” section, enter your conflict of interest statement in the “Confidential to Editor” section, and submit your "Accept" recommendation.

Reviewer #1: All comments have been addressed

Reviewer #2: All comments have been addressed

2. Is the manuscript technically sound, and do the data support the conclusions?

Reviewer #1: Yes

Reviewer #2: Yes

3. Has the statistical analysis been performed appropriately and rigorously?

Reviewer #1: Yes

Reviewer #2: Yes

4. Have the authors made all data underlying the findings in their manuscript fully available?

Reviewer #1: Yes

Reviewer #2: Yes

5. Is the manuscript presented in an intelligible fashion and written in standard English?

Reviewer #1: Yes

Reviewer #2: (No Response)

6. Review Comments to the Author

Reviewer #1: The authors took the reviews seriously and answered in a satisfactory way all questions and requests. As such, it is my opinion that the paper should be accepted as is.

Reviewer #2: As Reviewer #2, I have reviewed the revised manuscript (PONE-D-25-22333R1) and your responses to my initial comments. Overall, I am pleased with the improvements.

The revisions have improved scientific rigor and clarity. However, consider adding baseline t-SNE plots (without CRAE) for comparison and exploring adaptive thresholds in future work. With these enhancements, the manuscript is now suitable for publication. Well done!

7. PLOS authors have the option to publish the peer review history of their article (what does this mean?). If published, this will include your full peer review and any attached files.

Reviewer #1: No

Reviewer #2: No

---

## [Editor Report · Acceptance letter]

PONE-D-25-22333R1

PLOS ONE

Dear Dr. Li,

I'm pleased to inform you that your manuscript has been deemed suitable for publication in PLOS ONE. Congratulations! Your manuscript is now being handed over to our production team.

Kind regards,

on behalf of

Professor Haofeng Zhang

Academic Editor

PLOS ONE